# Seeing Before Reasoning: A Unified Framework for Generalizable and Explainable Fake Image Detection

## Abstract

Detecting AI-generated images with multimodal large language models (MLLMs) has gained increasing attention, due to their rich world knowledge, common-sense reasoning, and potential for explainability. However, naively applying those MLLMs for detection often leads to suboptimal performance. We argue that the root of this failure lies in a fundamental mismatch: *MLLMs are asked to reason about fakes before they can truly see them.* First, **they do not really see**: existing MLLMs' vision encoders are primarily optimized for semantic-oriented recognition rather than the perception of low-level signals, leaving them insensitive to subtle forgery traces. Without access to reliable perceptual evidence, the model grounds its judgment on incomplete and limited visual observations. Second, existing finetuning data for detection typically uses narrow, instruction-style formats, which diverge sharply from the diverse, heterogeneous distributions seen in pretraining. In the absence of meaningful visual cues, the model therefore exploits these linguistic shortcuts, resulting in catastrophic forgetting of pretrained knowledge (even the basic dialogue capabilities). In response, we advocate for a new paradigm: *seeing before reasoning*. We propose that MLLMs should first be trained to perceive artifacts—strengthening their artifact-aware visual perception—so that subsequent reasoning is grounded in actual observations. We therefore propose **Forensic-Chat**, a generalizable, explainable, and still-conversational (for multi-round dialogue) assistant for fake image detection. Specifically, we first refine the vision encoder only via self-reconstruction while freezing the LLM, sensitizing it to artifacts without sacrificing pretrained knowledge (Stage 1). Then, we construct a multi-round dialogue finetuning data for detection, which is designed to progressively guide the model from artifact perception to common-sense reflection, enabling dialectical reasoning about *why an image is fake* and *what a real version should look like* (Stage 2). We also propose **ExplainFake-Bench**, a benchmark tailored for the evaluation of the MLLM's explainability for image forensics from five key aspects. Extensive experiments show the superiority of generalization and genuinely reliable explainability.

## 1 Introduction

The rapid proliferation of AI-generated images (AIGIs) has intensified concerns about image authenticity, fraud, and copyright violations (Goodfellow et al., 2014; Ho et al., 2020; Rombach et al., 2022; Yan et al., 2025b;a). Multimodal Large Language Models (MLLMs) are appealing candidates for detection as they couple strong visual understanding with language-based semantic reasoning and human-understandable explanation (Wu et al., 2023). However, directly applying MLLMs to detection tasks often results in suboptimal performance—frequently underperforming compared to conventional detectors (Jia et al., 2024). Prior research attributes this shortcoming to an inherent limitation in MLLMs: a lack of fine-grained visual perception capabilities needed to capture low-level artifacts, leading to the use of external detectors to compensate (Chen et al., 2024b; Zhou et al., 2025b). This inevitably risks introducing another shortcut: the MLLM may simply mimic the expert's predictions rather than genuinely perceiving the subtal artifacts by itself.

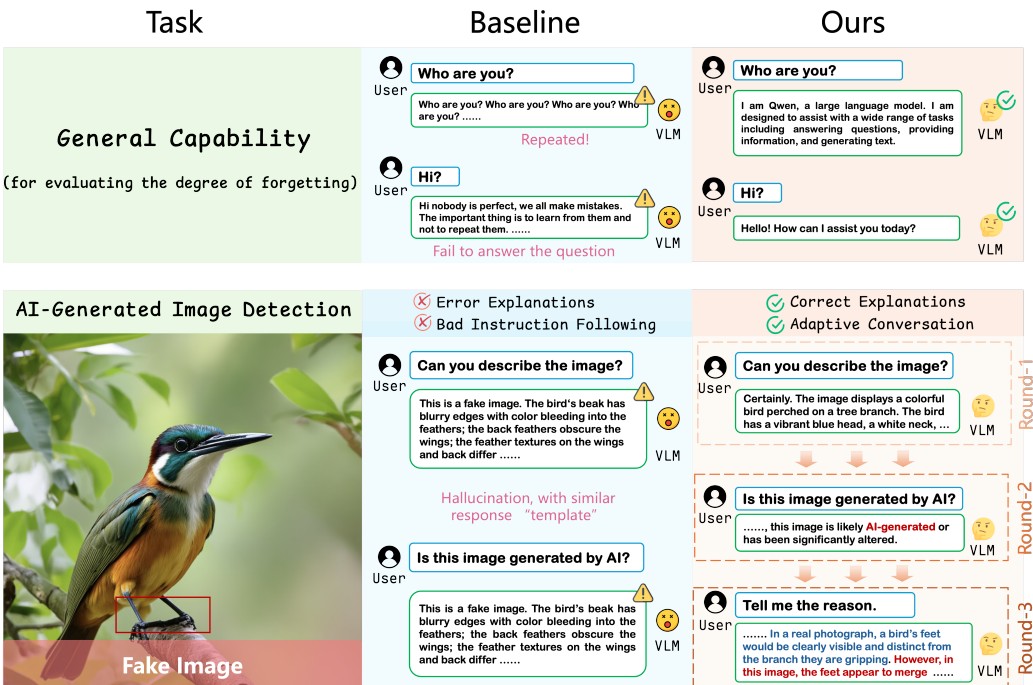

Figure 1: Illustration of a key limitation of existing MLLM-based detectors: models trained specifically for detection **fail to follow even basic instructions**. Moreover, the baseline **produces nearly identical responses across different questions**, even those unrelated to forensics. This undermines the reliability of the MLLM's explanations, as they lack fundamental instruction-following capabilities. In contrast, our proposed method supports conversational multi-round interaction and provides more consistent, trustworthy explanations to the users while achieving SOTA performance in generalization and robustness.

In this work, we argue that the core issue lies in a critical mismatch: *Existing MLLMs are trained to reason about fakes before they can truly see them.* **First**, the vision encoders in MLLMs are originally optimized for high-level semantic alignment with language, leaving them insensitive to the low-level signals and resulting in the MLLMs cannot really "see" the subtal (but potentially generalizable) forensic artifacts. **Second**, existing fine-tuning strategies (Kang et al., 2025; Wen et al., 2025) typically lean on narrow, instruction-style supervision (e.g., repeated Q/A templates of the form "Is this image real or fake?" / "This image is fake because. . . "), which creates a difficulty mismatch—simple prompts paired with elaborate answers—that encourages *shortcut learning*: the model maps superficial visual cues to verbal templates rather than developing robust causal reasoning about forgeries.

As shown in Fig. 1, in the most basic diagnostic task, the baseline model (Wen et al., 2025) fails to comprehend the input, generating responses that are entirely irrelevant to the question. More concerning, in the forensic detection task—even when explicitly instructed only to *describe* the image (with no mention of authenticity or manipulation)—the model still produces forensic-oriented outputs, rather than truly following the instruction. This behavior significantly undermines the confidence in the model's **explainability**: *if an MLLM cannot even perform basic diagnosis or follow the so-simple instructions, its generated explanations cannot be trusted as accurate reflections of the input.* Additional to the explainability, prior research also indicates that the severe catastrophic forgetting of pretrained knowledge can also largely hurt the model's generalization toward unknown forgery methods (Yan et al., 2024b).

To address these issues, we argue that, a viable detector must (i) have strong artifact-aware visual perception without sacrificing the model's linguistic competence; (i) instill a *general* and *dialectical* reasoning pattern which resists template memorization and considers plausible real counterparts. We therefore present **Forensic-Chat**, a generalizable, explainable, and still-conversational assistant for AIGI detection, which explicitly targets both requirements. We begin with a **Visual**

**Enhancement (VE)** stage that sensitizes the vision encoder in the MLLM to subtle traces by leveraging self-reconstruction images (aligning semantics, focusing low-level cues) for fine-tuning, while *freezing* the LLM. This preserves the MLLM's original pre-trained knowledge and semantic priors, yet improves model's perception capability to the low-level artifacts. Building on this, we further introduce **Dialectical Fine-Tuning (DFT)**, a multi-turn data curriculum that progresses from basic image understanding to fake-trace perception and common-sense reflection. By requiring the model to reason about *why an image is fake* and *what a plausible real counterpart would look like*, DFT suppresses shortcut solutions and promotes counterfactual, stepwise analysis.

To comprehensively evaluate the effectiveness of our method, we test our method from **three different yet critical dimensionalities**, including (1) the generalization of detection, (2) the reliability and accuracy of the output explanations, and (3) capability preservation of pretrained knowledge. **For generalization**, our method achieves state-of-the-art (SOTA) performance across multiple AIGI-detection benchmarks, such as the standard GenImage (Zhu et al., 2023) benchmark, and the recently released GenImage++ (Zhou et al., 2025a), AIGI-Holmes (Zhou et al., 2025b), etc, demonstrating its strong detection capability toward the latest and unknown generators. **For explainability**, we propose a new comprehensive benchmark called **ExplainFake-Bench**, specifically to evaluate the explainability of the MLLM's output for detection, covering correctness, evidence sufficiency, instruction following, factual accuracy, and etc. Results show that our method can provide a notably more reliable explanations than other MLLM-based detectors. **For knowledge preservation** (general capabilities), we evaluate our method with the baseline MLLM (without additional fine-tuning for AIGI detection) and other MLLM-based detectors on the widely-used benchmarks for general MLLM evaluation (Yue et al., 2024; Ying et al., 2024; Fu et al., 2024a; x AI, 2024). Results demonstrate our proposed training strategy can maximually preserve the pretrained knowledge and maintain its basic dialogue capabilities, leading to a more trustworthy and reliable explanations.

Our contributions are summarized below.

- In this work, **we first propose an entirely new paradigm** for addressing the suboptimal performance of using MLLM for AIGI detection. The key is **"*seeing before reasoning*"**, where the model should first percept the artifacts so that the reasoning process is truly based on the seen cues. This leads to a generalizable and explainable detection result.
- Following this principle, we propose a **pure MLLM-based framework** with strong detection performance, reliable explainability, and still-strong conversational capabilities. We implement this by (1) encouraging the *learning of artifact-aware visual perception*, and (2) *dialectical thinking by applying commonsense reasoning and reflection*, finally achieving state-of-the-art (SOTA) performance across different benchmarks.
- We also provide a new comprehensive benchmark, namely **ExplainFake-Bench**, tailored to **quantify the explainability** of the MLLM's output from five key perspectives. We demonstrate a significant improvement of our method over other MLLM-based baselines in terms of explainability.
- Our **technical contributions** include: (1) *a simple yet effective strategy to enhance the visual encoder's sensitivity to forensic artifacts* by fine-tuning only this component, while keeping the LLM frozen to preserve its pretrained linguistic knowledge; (2) a novel *dialectical reasoning strategy* that leverages multi-turn dialogue to elicit commonsense inference, enabling the model to contrast the input with plausible real-world counterparts for more comprehensive reasoning.

## 2 RELATED WORK AND MOTIVATION

**Traditional AIGI Detection**   Early work, CNNSpot (Wang et al., 2020), trains a standard CNN to detect AI-generated images (AIGI), showing that while such detectors perform well on images from known generators, they often fail to generalize to unseen ones. UnivFD (Ojha et al., 2023) improves generalization by using CLIP as a backbone, leveraging the strong representational power of pretrained vision models. Follow-up methods (Liu et al., 2024; Tan et al., 2024a; Zheng et al., 2024; Yan et al., 2024b) explore advanced model architectures and image preprocessing strategies to further boost performance across diverse generators. For instance, C2P-CLIP (Tan et al., 2024a) enhances the pretrained CLIP model by explicitly embedding "real" and "fake" semantic concepts into its learning process. Other approaches (Tan et al., 2024b; Chu et al., 2025; Li et al., 2024; Karageorgiou et al., 2025) focus on frequency-domain artifacts, demonstrating that signals in the frequency domain are highly effective for distinguishing real from synthetic images. Additionally, NPR (Tan et al., 2024c) examines upsampling artifacts—distinctive traces introduced during the

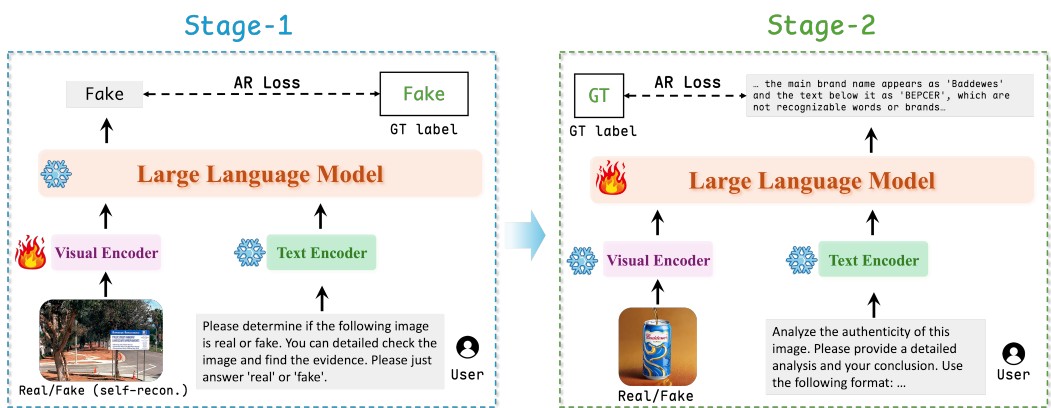

Figure 2: The overall pipeline of our method. In Stage 1, we exclusively fine-tune the parameters of the Vision Encoder, while in the subsequent stages, we only optimize the LLM.

image generation process—as reliable indicators of synthetic content. However, these conventional detection methods still suffer from two key limitations:

- *Explainability: these methods provide only a binary classification result, without offering any rationale for why an image is deemed real or fake.*

- *Robustness: they rely heavily on low-level, pixel-level artifacts, which are easily destroyed by common perturbations such as compression.*

**MLLM-based AIGI Detection** Instruction Fine-Tuning (IFT) is effective for teaching response formats but has limited capacity to inject new factual knowledge (Ren et al., 2024; Zhou et al., 2023; Lampinen et al., 2025). As a result, MLLMs can leverage their pre-existing knowledge but often fail on tasks requiring novel or fine-grained facts. Existing MLLM-based detection methods (Kang et al., 2025; Gao et al., 2025; Wen et al., 2025; Lin et al., 2025; He et al., 2025) overlook this limitation, leading to two critical design flaws. **First**, their fine-tuning paradigm encourages shortcut learning, which use training data with simple prompts (e.g., "Is this image real or fake?") paired with long, detailed answers—a mismatch that overwhelms the model's reasoning capacity. Instead of learning to analyze images causally, the MLLM learns shallow mappings from visual artifacts to textual templates. The model appears to reason but merely memorizes patterns, failing to develop reliable forgery detection capabilities—and risking degradation of its pretrained knowledge in the process. **Second**, they inadequately address the weak visual perception of MLLMs. Some methods co-fine-tune both the vision encoder and the LLM (Zhang et al., 2025), which can disrupt the LLM's internal knowledge and alignment. Others bypass the issue entirely by integrating external expert detectors (Chen et al., 2024b; Zhou et al., 2025b; Peng et al., 2025). However, this introduces another shortcut: the MLLM learns to copy the expert's predictions rather than performing genuine visual analysis. In short, previous works fail to solve two main problems:

- *How to use a MLLM's pre-trained knowledge to detect fake images without damaging it.*

- *How to strengthen the MLLM's visual encoder to percept forgery artifacts.*

To this end, our work focuses on addressing these two problems. We aim to enhance the MLLM's perception capability to detect fake images, specifically for image forensics, while preserving its pre-trained knowledge.

## 3 METHOD

We propose a two-stage training strategy: **1)** Visual Enhancement (VE) and **2)** Dialectical Fine-Tuning (DFT). The two stages build upon supervised fine-tuning: VE first enhances the model's visual perception, then DFT improves its reasoning based on high-level knowledge. These stages realize the principle of *"seeing before reasoning"*, where reliable perception serves as the foundation for subsequent reasoning. The overall pipeline is illustrated in Figure 2.

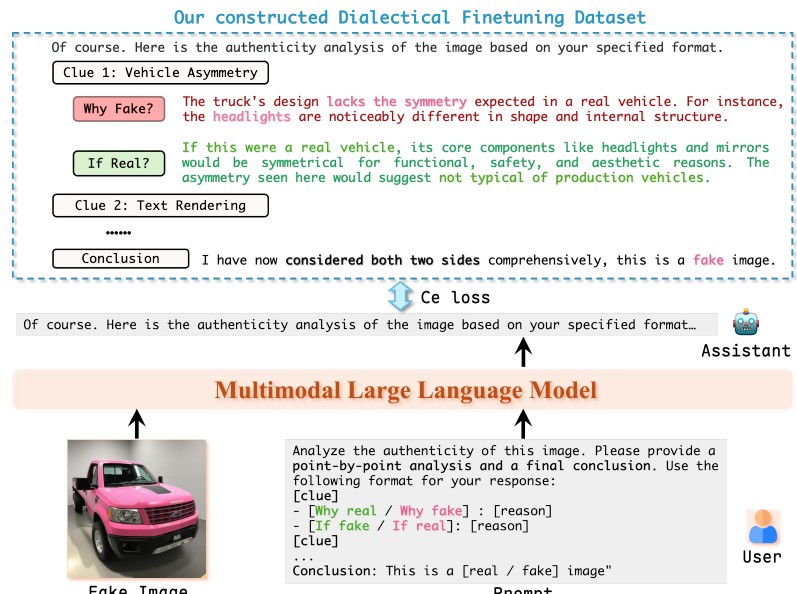

Figure 3: The detailed illustration of the **Stage 2** of our framework, where we first introduce a **dialectical finetuning strategy** that contrasts externally detected fake clues with internal common-sense and world knowledge. By weighing conflicting signals, the model enhances robustness against deception while preserving pretrained knowledge for reliable reasoning.

## 3.1 Two-stage Training Pipeline

**Visual Enhancement (VE)–Stage 1** While pre-trained MLLMs excel at recognizing high-level semantic content, they often lack the specialized capacity to perceive low-level forgery artifacts. To address this gap, we propose a fine-tuning method to cultivate this perceptual ability directly within the MLLM, thereby unifying detection in a single, cohesive model. To do this, we construct a training dataset, $\mathcal{P}_1$, containing fake images that exhibit only low-level artifacts, while excluding obvious semantic errors like an anomalous number of fingers. To decouple perception from reasoning and avoid any external classification head, we supervise the system using the LLM itself. During this stage, we freeze the LLM's parameters and fine-tune only the visual encoder. We cast detection as a simple question–answering task and optimize the native autoregressive loss so that the model produces the target token (e.g., "real" or "fake"). Updating only the ViT while keeping the LLM frozen isolates the learning of visual perception in the encoder and preserves the LLM's world knowledge and high-level reasoning. This strategy aims to improve the model's low-level perception without disrupting its pre-trained semantic knowledge.

**Dialectical Fine-Tuning (DFT)–Stage 2** A major bottleneck in detecting *semantic* artifacts is data scarcity: annotated sets are small and cover only a narrow range of forgeries, which makes models prone to shortcut learning and overfitting. DFT leverages the MLLM's rich pretrained knowledge to build a *reasoning* capability that is robust to such shortcuts. The core idea is to judge authenticity by checking for contradictions between visual evidence ("what it sees") and internal world knowledge ("what it knows"). To support this, we construct a semantic-artifact dataset $\mathcal{P}_2$ where each image is annotated with (i) a commonsense rule and (ii) a description of the visual evidence that may support or violate that rule. An example data is shown in Figure 3. During DFT we fine-tune only the LLM, keeping the visual encoder frozen, so the model learns *how to reason about what it perceives* without altering the visual features learned in Stage 1. Single-turn, fixed-format instruction tuning—common in prior work (Kang et al., 2025; Gao et al., 2025; Wen et al., 2025)—encourages fitting output templates rather than the *key information* for detection. It maps a simple question to a complex answer and yields a training distribution misaligned with the model's autoregressive pretraining, which harms generalization. We therefore transform each seed annotation $\mathcal{S}$ into a *multi-turn dialogue* that decomposes the task into progressive steps. Each prompt requests a deeper analysis conditioned on the prior context, forming a gentle curriculum that aligns better with how

the model was pretrained (as analyzed in subsection D of the Appendix). This conversational data construction aligns the joint distribution of the training data with the model's inherent knowledge base. It successfully disentangles the challenge of learning *what* to reason about (the semantics) from *how* to present the answer (the format). This ensures the model learns the crucial underlying logic for the task while retaining its ability to follow explicit formatting commands.

**Optimization** In this paper, we eventually utilize the autoregressive loss to optimize the MLLMs, where the loss function $\mathcal{L}(\theta)$ can be formulated as

$$\mathcal{L}(\theta) = -\sum_{i=1}^{n} \log P(x_i|x_1, x_2, \ldots, x_{i-1}; \theta),\tag{1}$$

where $(x_1, x_2, \ldots, x_{i-1})$ is the context tokens, and $\theta$ represents the model parameters.

### 3.2 DATA CONSTRUCTION

**Dataset $\mathcal{P}_1$ in Stage 1** Large-scale pre-training has rendered Multimodal Large Language Models (MLLMs) highly proficient in perceiving high-level image semantics. A significant limitation, however, is their inability to detect low-level artifacts. The subtle artifacts are invisible to the human eye but important for the development of generalizable AIGI detectors. Instead of using standard AI-generated images, we create training pairs consisting of a real image and its reconstruction, $(I_{real}, I_{recon})$. We use a pre-trained VAE decoder to generate image reconstructions, which serve as our "pseudo-fake" data. This process removes high-level semantic artifacts while introducing low-level artifacts from the VAE. As a result, the images in each pair are semantically almost identical; the only meaningful difference for the model to learn is the presence of these artifacts. This approach forces the MLLM to learn low-level architectural traces rather than semantic cues, leading to a model that generalizes much better. We will provide more details in the Appendix E.

**Dataset $\mathcal{P}_2$ in Stage 2** To address shortcut learning, where models often overfit to superficial templates, we constructed a new dataset, $\mathcal{P}_2$, designed to foster commonsense-based reasoning. The construction process involves two primary stages: dialectical seed annotation and multi-turn dialogue generation. First, we perform dialectical annotation to create a set of core reasoning seeds. For each image in our source pool, we generate a contrastive pair of statements: one describing the visual evidence and another representing a corresponding commonsense rule. This is achieved using a reverse operation where a powerful LLM transforms descriptions of anomalies into their plausible, real-world counterparts (and vice-versa), resulting in contradictory annotation pairs for both real and fake images. Second, to create a data format that aligns with MLLM pre-training and avoids incentivizing format-overfitting, we expand these seed annotations into multi-turn dialogues. Each conversational turn progressively guides the model through the reasoning process, breaking down a complex judgment into simpler and sequential steps. The final dataset, $\mathcal{P}_2$, is therefore composed of multi-turn conversational samples. This structure is designed to explicitly disentangle the core reasoning task (identifying a logical contradiction) from the challenge of learning a specific output format. We will provide more details in the Appendix E.

## 4 EXPERIMENTS

In this section, we conduct extensive experiments to comprehensively evaluate the generalization performance of our method on detection tasks. We also present detailed ablation studies and provide several insightful analyses. The experimental setting is introduced in the Appendix C

**Evaluation Metric and Comparison Methods** In evaluation, following the previous works, we utilize the macro accuracy as the metrics. To comprehensively verify the effectiveness of our method, we also provide the results of the following models: Xception (Chollet, 2017), CNNSpot (Wang et al., 2020), F3Net (Qian et al., 2020), GramNet (Liu et al., 2020), UniFD (Ojha et al., 2023), NPR (Tan et al., 2024c), AIDE (Yan et al., 2024a), DIRE (Wang et al., 2023), DRCT (Chen et al., 2024a), OMAT (Zhou et al., 2025a), AIGI-Holmes (Zhou et al., 2025b), SAFE (Li et al., 2025a), C2P-CLIP (Tan et al., 2025), FatFormer (Liu et al., 2024), CO-SPY (Cheng et al., 2025).

### 4.1 GENERALIZATION RESULTS

**Performance on the AIGI detection benchmarks** To assess the effectiveness and generalization capabilities of our approach, we conduct a comparative evaluation against prior AIGI detectors on

Table 1: The generalizable performance (ACC (%)) in GenImage Dataset (Zhu et al., 2023).

| Model | MidJourney | SDv1.4 | SDv1.5 | ADM | GLIDE | Wukong | VQDM | BigGAN | AVG |
|---|---|---|---|---|---|---|---|---|---|
| Xception | 57.97 | 98.06 | 97.98 | 51.16 | 57.51 | 97.79 | 50.34 | 48.74 | 69.94 |
| CNNSpot | 61.25 | 98.13 | 97.54 | 51.50 | 55.13 | 93.51 | 51.83 | 51.06 | 69.99 |
| F3Net | 52.26 | 99.30 | 99.21 | 49.64 | 50.46 | 98.70 | 45.56 | 49.59 | 68.09 |
| GramNet | 63.00 | 94.19 | 94.22 | 48.69 | 46.19 | 93.79 | 49.20 | 44.71 | 66.75 |
| UniFD | 77.29 | 97.01 | 96.67 | 50.94 | 78.47 | 91.52 | 65.72 | 55.91 | 77.29 |
| NPR | 62.00 | **99.75** | 99.64 | 56.79 | 82.69 | 97.89 | 54.43 | 52.26 | 75.68 |
| AIDE | 79.38 | 99.74 | **99.76** | 78.54 | 91.82 | 98.65 | 80.26 | 66.89 | 86.88 |
| DIRE | 51.11 | 55.07 | 55.31 | 49.93 | 50.02 | 53.71 | 49.87 | 49.85 | 51.86 |
| DRCT/Conv-B | **94.43** | 99.37 | 99.19 | 66.42 | 73.31 | **99.25** | 76.85 | 59.41 | 83.53 |
| DRCT/UniFD | 85.82 | 92.33 | 91.87 | 75.18 | 87.44 | 92.23 | 89.12 | 87.38 | 87.67 |
| OMAT | 90.36 | 97.52 | 97.46 | 83.82 | 97.41 | 97.62 | 95.53 | 97.34 | 94.63 |
| **Forensic-Chat** | 93.20 | 99.10 | 98.85 | **96.12** | **98.70** | 98.14 | **97.82** | **98.49** | **97.55** |

Table 2: The generalizable performance (ACC (%)) in GenImage++ Dataset (Zhou et al., 2025a). This dataset includes fake images only.

| Model | Flux | Flux Multi | Flux Photo | Flux Real | SD1.5 Multi | SDXL Multi | SD3 | SD3 Photo | SD3 Real | AVG |
|---|---|---|---|---|---|---|---|---|---|---|
| Xception | 36.86 | 10.48 | 4.65 | 5.45 | 97.27 | 20.63 | 38.00 | 5.83 | 15.06 | 26.03 |
| CNNSpot | 37.38 | 6.89 | 8.71 | 5.28 | 84.41 | 34.79 | 47.70 | 7.48 | 25.55 | 28.69 |
| F3Net | 25.18 | 7.79 | 2.83 | 7.90 | 94.15 | 24.01 | 46.67 | 0.84 | 30.28 | 26.63 |
| GramNet | 37.83 | 16.71 | 8.01 | 19.71 | 96.49 | 28.65 | 48.55 | 8.33 | 55.71 | 35.55 |
| NPR | 35.38 | 13.19 | 8.48 | 19.41 | 93.63 | 15.40 | 32.38 | 12.45 | 27.58 | 28.66 |
| SPSL | 67.13 | 16.55 | 43.76 | 25.73 | 71.14 | 17.74 | 44.58 | 16.22 | 29.75 | 36.96 |
| SRM | 8.46 | 2.92 | 0.37 | 1.93 | 96.62 | 6.39 | 9.97 | 0.55 | 4.43 | 14.63 |
| DRCT/Conv-B | 73.02 | 51.91 | 54.72 | 66.40 | 100.00 | 77.19 | 79.10 | 82.93 | 76.58 | 73.54 |
| DRCT/UniFD | 71.08 | 63.97 | 46.83 | 62.42 | 99.19 | 64.84 | 72.28 | 70.70 | 73.55 | 69.43 |
| OMAT | 96.53 | 92.55 | 97.60 | 97.67 | 100.00 | 99.17 | 98.27 | 90.38 | 98.82 | 96.78 |
| **Forensic-Chat** | **99.58** | **97.15** | **99.98** | **99.97** | 93.96 | 90.23 | 97.88 | **98.49** | **99.78** | **97.44** |

three established benchmarks (Zhu et al., 2023; Zhou et al., 2025a;b). The results, summarized in Tables 1, 2, and 3, demonstrate that Forensic-Chat consistently achieves state-of-the-art performance. Notably, on the AIGI-Holmes benchmark (Table 3), our method significantly outperforms the MLLM-based AIGI-Holmes* across every image generator. This superiority is quantified by a 5.51 percentage point increase in average accuracy, from 92.30% to 97.81%. Such outstanding performance across multiple diverse benchmarks strongly validates the effectiveness of our proposed method. Further experimental results on additional benchmarks are provided in the Appendix

**Performance on Benchmarks in the Wild** In Table 4, we evaluate our method's robustness on the WildRF (Cavia et al., 2024) and AIGI-Bench (Li et al., 2025b) benchmarks. These datasets are specifically designed to reflect real-world conditions by incorporating distortions commonly found on social media, such as image compression. Across these challenging benchmarks, Forensic-Chat consistently outperforms competing methods. While many existing detectors perform well on standard benchmarks by identifying low-level artifacts, their effectiveness collapses when these artifacts are destroyed by compression. This performance degradation also occurs to some extent in our Stage 1 model, indicating that enhanced artifact perception alone is insufficient for robust detection. In contrast, the introduction of Stage 2 brings a significant performance improvement. We attribute this advantage to our second stage, which enables the model to move beyond a reliance on fragile, low-level artifacts and instead learn more robust features resilient to real-world distortions.

Table 3: The generalizable performance (ACC (%)) in AIGI-Holmes Dataset (Zhou et al., 2025b). We obtained the experimental results from the original paper. AIGI-Holmes* means the results using the MLLM only without further ensembling another external dedicated detector.

| Model | Janus | J-Pro-1B | J-Pro-7B | Show-o | LlamaGen | Infinity | VAR | PixArt-XL | SD3.5 L | FLUX | AVG |
|---|---|---|---|---|---|---|---|---|---|---|---|
| CNNSpot | 70.00 | 70.90 | 85.00 | 72.20 | 61.90 | 86.80 | 59.90 | 78.20 | 63.80 | 79.90 | 72.90 |
| AntiFakePrompt | 72.20 | 84.30 | 84.80 | 86.20 | 96.20 | 83.60 | 90.70 | 81.70 | 92.80 | 66.10 | 83.90 |
| UnivFD | 87.60 | 96.90 | 96.40 | 85.90 | 93.10 | 79.20 | 64.30 | 75.70 | 87.80 | 69.60 | 83.60 |
| NPR | 51.20 | 69.50 | 73.90 | 93.70 | 93.50 | 93.80 | 85.90 | 93.40 | 91.60 | 93.60 | 84.00 |
| LaRE | 70.80 | 74.70 | 95.60 | 80.00 | 91.60 | 77.90 | 98.80 | 82.20 | 94.10 | 84.30 | 85.00 |
| RINE | 89.90 | 98.70 | 97.20 | 98.80 | 99.10 | 99.20 | 85.00 | 98.90 | 97.80 | 97.10 | 96.20 |
| AIDE | 91.20 | **98.90** | **97.80** | 98.00 | 99.40 | 98.70 | 93.60 | 98.60 | **99.40** | 94.40 | 97.00 |
| AIGI-Holmes* | 80.20 | 91.90 | 89.60 | 98.00 | 98.00 | 98.40 | 76.00 | 98.50 | 97.80 | 94.20 | 92.30 |
| **Forensic-Chat** | **92.19** | 93.18 | 97.51 | **99.70** | **99.66** | **99.88** | **99.09** | **99.66** | 97.85 | **99.40** | **97.81** |

Table 4: Performance (ACC (%)) on WildRF (Cavia et al., 2024) and AIGI-Bench (Li et al., 2025b), two benchmarks designed to reflect real-world conditions and evaluate the robustness of detectors.

| Method | WildRF | | | | AIGI Bench | | |
|---|---|---|---|---|---|---|---|
| | FaceBook | Reddit | Twitter | **AVG** | SocialRF | CommunityAI | **AVG** |
| FatFormer | 64.38 | 76.65 | 40.00 | 60.34 | 57.98 | 50.62 | 54.30 |
| CO-SPY | 50.00 | 56.79 | 73.30 | 60.03 | 55.54 | 53.02 | 54.28 |
| C2P-CLIP | 54.38 | 68.40 | 47.27 | 56.68 | 53.13 | 50.98 | 52.06 |
| SAFE | 62.50 | 61.70 | 40.33 | 54.84 | 58.00 | 54.25 | 56.13 |
| AIDE | 75.00 | 55.48 | 48.00 | 59.49 | 57.80 | 54.15 | 55.98 |
| **Ours (Only Stage1)** | 51.88 | 56.00 | 76.80 | 61.56 | 57.50 | 62.32 | 59.91 |
| **Ours (Stage 1 + Stage 2)** | **77.81** | **83.37** | **82.07** | **81.08** | **74.57** | **89.81** | **82.19** |

Table 5: Ablation study of our methods for different stages (ACC (%)).

| Variant | GenImage | | | GenImage++ | | | | AVG |
|---|---|---|---|---|---|---|---|---|
| | MidJourney | BigGAN | SDv1.5 | Flux | Flux Real | SD3 | SD3 Real | |
| Baseline | 50.19 | 49.98 | 50.09 | 1.70 | 5.83 | 0.33 | 2.12 | 22.89 |
| Stage 1 | 82.88 | 95.75 | 95.95 | 87.15 | 87.15 | 87.43 | 95.65 | 90.28 |
| Stage 2 | 82.57 | 88.71 | 88.31 | 98.25 | **99.98** | 95.50 | **99.63** | 93.28 |
| Stage 1 + Stage 2 | **93.20** | **98.49** | **98.85** | **99.58** | 99.97 | **97.88** | **99.78** | **98.25** |

**Ablation Study** We conducted an ablation on Qwen-2.5-VL-7B with different training strategies. As summarized in Table 5, the proposed two-stage pipeline yields substantial gains. The unfine-tuned baseline shows little ability (AVG 22.89%). After Stage 1 (VE), the average accuracy jumps to 90.28%. Applying Stage 2 (DFT) further lifts performance to 93.28% ( we remove all Flux images during Stage 2 training to ensure fairness). Stage 1, which strengthens artifact-aware perception, is especially effective on traditionally generated images (e.g., BigGAN, SDv1.5), whereas Stage 2 better handles modern generators (e.g., Flux, SD3). Combining both stages delivers the best overall result (97.23% AVG), validating the effectiveness of our seeing-before-reasoning framework.

## 4.2 EXPLAINABILITY RESULTS

**ExplainFake-Bench** To assess the quality of judgments, we employ an LLM-as-Judge framework and introduce an explainable benchmark, *ExplainFake Bench*. We evaluate explainability across five dimensions: **Correctness** (alignment of the final judgment with the ground truth), **Specificity** (use of concrete, identifiable details), **Logical Consistency** (coherence of the reasoning), **Factual Accuracy** (consistency of the explanation with the visual content), and **Instruction Following** (adherence to the required format). *Notably, a significant penalty will be applied to the ratings if the model's final judgment is incorrect.* Further details on our evaluation methodology are provided in the Appendix E.3. The results of Table 6 highlight a critical distinction between general-purpose MLLMs and specialized AIGI detectors. Although Gemini-2.5-pro achieves the highest average rating among commercial MLLMs, its performance on the **Correctness** dimension is surpassed by MLLM-based AIGI detectors. This suggests a limitation in the inherent capability of general models for the nuanced task of AIGI detection. While specialized detectors demonstrate strong detection capabilities, reflected in their high **Correctness** scores, they often suffer from a significant trade-off. For instance, FakeVLM excels in accuracy but fails markedly across the other four dimensions: Specificity, Logical Consistency, Factual Accuracy, and Instruction Following. Its poor performance, with scores averaging around 2, indicates that it may have degraded into a simple binary classifier, sacrificing its world knowledge and reasoning abilities for mere classification accuracy. In contrast, our proposed model, Forensic-Chat, shows substantial improvements across all dimensions. The first stage of our method alone, which enhances the visual encoder through a decoupled artifact-aware perception mechanism, yields significant gains, particularly in **Specificity** and **Instruction Following**. For the Forensic-Chat, Stage 1 not only improves the detection but also preserves the model's pre-trained knowledge, making the clarity and structure of the explanation better. Furthermore, applying our full two-stage framework, which incorporates Domain-Following Tuning (DFT), further elevates performance. This indicates that DFT enables the MLLM to adopt a more robust reasoning process, leveraging its internal world knowledge rather than merely overfit-ting to the training data distribution. ***Consequently, Forensic-Chat achieves the best average rating***

*on ExplainFake Bench, proving that our two-stage framework is highly effective in generating accurate and logically consistent judgments for real-world scenarios of AIGI detection.*

### 4.3 RESULTS OF GENERAL KNOWLEDGE PRESERVATION

**General Understanding Benchmark** To evaluate the preservation of pre-trained knowledge, we test our model on several general-purpose multimodal understanding benchmarks that probe fundamental capabilities such as perception, world knowledge, and commonsense reasoning (x AI, 2024; Ying et al., 2024; Fu et al., 2024b;a). The results, shown in Table 7, reveal a stark contrast in how different models handle the integration of the new detection task. FakeVLM suffers from severe catastrophic forgetting; the process of fine-tuning for AIGI detection significantly degrades its pre-trained knowledge. This outcome is counterproductive to the primary goal of leveraging the rich knowledge inherent in MLLMs to generalize to unseen data. In contrast, Forensic-Chat maintains robust performance across all general benchmarks, demonstrating its ability to effectively preserve foundational multimodal abilities. It highlights a key advantage of our method: Forensic-Chat can achieve SOTA performance in the specialized domain of AIGI detection while simultaneously retaining its broad, pre-trained knowledge. This successful balance underscores the efficacy of our framework in preventing catastrophic forgetting.

Table 6: Evaluation of the explainability of the MLLM-based detector on *ExplainFake-Bench*, assessed using the LLM-as-Judge (Zheng et al., 2023). GPT-4o (OpenAI, 2025) was selected as the evaluator. The models with the superscript [*] are commercial MLLMs.

| Model | Correctness | Specificity | Logical Consistency | Factual Accuracy | Instruction Following | **AVG** |
|---|---|---|---|---|---|---|
| **General MLLM** | | | | | | |
| GPT-4o[*] | 3.1285 | 2.9358 | 3.2393 | 3.2015 | 3.2469 | 3.1504 |
| Gemini-2.5-Pro[*] | 3.8722 | 4.0326 | 4.1529 | 4.0313 | 4.1629 | 4.0504 |
| Claude-Sonnet-4[*] | 3.3277 | 3.3199 | 3.4807 | 3.3168 | 3.4652 | 3.3821 |
| Qwen-2.5-VL-7B | 3.0624 | 2.9358 | 3.1853 | 3.1339 | 3.2110 | 3.1057 |
| **MLLM-based AIGI Detector** | | | | | | |
| FakeVLM | 4.0915 | 1.6905 | 2.6529 | 2.3446 | 2.4975 | 2.6554 |
| **Ours (Stage 1)** | 4.4577 | 3.7581 | 4.3842 | 3.6204 | 4.4528 | 4.1346 |
| **Ours (Stage 1 + Stage 2)** | 4.5363 | 3.9461 | 4.4048 | 3.8158 | 4.4236 | **4.2253** |

Table 7: Comparison of our model and baselines on general multimodal understanding benchmarks. All models were evaluated using VLMEvalKit (Duan et al., 2024) and Ms-Swift (Zhao et al., 2024).

| Method | BLINK | RealWorldVQA | MME | MMT-Bench$_{VAL}$ |
|---|---|---|---|---|
| **General MLLM** | | | | |
| Qwen-2.5-VL-3B | 0.4750 | 0.6588 | 1590 | 0.6025 |
| Qwen-2.5-VL-7B | 0.5481 | 0.6758 | 1677 | 0.5948 |
| LLaVA-1.5-7B | 0.4171 | 0.5424 | 1436 | 0.4713 |
| **MLLM-based AIGI Detector** | | | | |
| FakeVLM | 0.3761 | 0.5385 | 1221 | 0.4445 |
| **Forensic-Chat** | 0.5139 | 0.6745 | 1625 | 0.5849 |

## 5 CONCLUSION

We introduce **Forensic-Chat**, a "*seeing before reasoning*" framework for AI-generated image detection that first enhances the MLLM's fine-grained perception capabilities and then finetuning it using the carefully-constructed dataset based on dialectical reasoning. The design couples artifact-aware visual perception with dialectical reasoning, leading to a unified, single, MLLM-based detector, without any using external detectors. Extensive results on standard, latest, and in-the-wild benchmarks verify consistent SOTA-level performance of our method in terms of generalization, explainability, robustness, and general knowledge preservation.

**Content Structure of the Appendix** Due to page constraints, we include additional analyses and experiments in the Appendix D, containing comprehensive ablation studies, more visual examples of model explanations, details of dataset construction, statement to USAGE OF LLM and Reproducibility Statement (Appendix A and B). For further details, please refer to the Appendix.

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

APPENDIX

## A  USAGE OF LLM

In this paper, large language models (LLMs), specifically Gemini-2.5-pro and GPT-4o, were used **exclusively for writing refinement**. They did not contribute to the research design, data analysis, or interpretation of results.

## B  REPRODUCIBILITY STATEMENT

To ensure the reproducibility of our findings, we have made significant efforts to provide comprehensive details of our methodology and experiments. All datasets used in this work are publicly available benchmarks. The specific construction of training data is detailed in Appendix E. Our proposed model, Forensic-Chat, is described in detail with its motivation (Section 1) and pipeline (Section 3). Key implementation details for training are provided in Appendix C. All backbone models were loaded from the official Hugging Face Hub checkpoints. To facilitate direct replication of our results, we will make our complete source code, pre-trained model weights, and experiment configurations publicly available upon publication.

## C  EXPERIMENT SETTING

**Implementation Details.**  We train Forensic-Chat using a two-stage fine-tuning with LoRA (Hu et al., 2022) adaptation. The Qwen-2.5-VL-7B (Bai et al., 2025) was selected as the backbone. The LoRA ranks are set to 16 and 128 for the three stages, respectively. As detailed in our methodology, the application of LoRA modules is stage-specific: they are initially applied only to the visual encoder, and subsequently to the large language model components in the later stages. The model is optimized using Adam (Kingma & Ba, 2014) with an initial learning rate of 0.0001, which is decayed following a cosine schedule. We adjust the dimensions of each image to achieve a total pixel count of $1024\times1024$, while holding the original aspect ratio constant. Notably, our model is trained on the open-source framework MS-Swift (Zhao et al., 2024) with the version `3.8.0.dev`.

**Data Source.**  All of our training data, for all real images and part of fake images, are from open-source datasets. The real images are collected from MS-COCO (Lin et al., 2014), Unsplash (Unsplash, 2025), Hand-Body (Narasimhaswamy et al., 2022), and LAION (Schuhmann et al., 2022). Fake images are partially sourced from the training set of SynthScars (Kang et al., 2025) and GenImage Zhu et al. (2023), and Echo-4o (Ye et al., 2025). Moreover, additional fake images are generated using Flux (Labs, 2024).

## D  MORE EXPERIMENTS

**Coarser Visual Granularity of MLLM**  We observed a strong correlation between the input image resolution and the detection performance of models trained on our dataset $\mathcal{P}_1$ (Figure 4). Specifically, performance was exceptionally poor on small images, yet it improved significantly when these same images were simply enlarged via resizing. This finding led us to hypothesize that the primary bottleneck is not the information content of the image itself, but rather the coarse granularity at which modern Multimodal Large Language Models (MLLMs) perceive visual details, a departure from traditional models like CLIP-ViT-Large-14 (Radford et al., 2021). We attribute this limitation to the inherent design of MLLM visual encoders and a pre-training focus on high-level semantic information. For instance, each visual token in Qwen2.5-VL-7B corresponds to a large $28\times28$ pixel area, which struggles to capture fine-grained artifacts. By enlarging the input, we force each coarse visual token to represent a smaller, more detailed patch of the original scene. Since performance is boosted without adding any new information to the image, this experiment confirms that coarse visual granularity is a key bottleneck of the MLLMs. This suggests that developing MLLM visual encoders with finer perceptual granularity is a promising direction for future work.

**Multi-Turn Dialogues**  We argue that the single-turn, question-answering data format common in AIGI detection poses a significant learning challenge for MLLMs. The mapping from a simple query to a complex answer is inconsistent with their pre-trained autoregressive nature. To validate

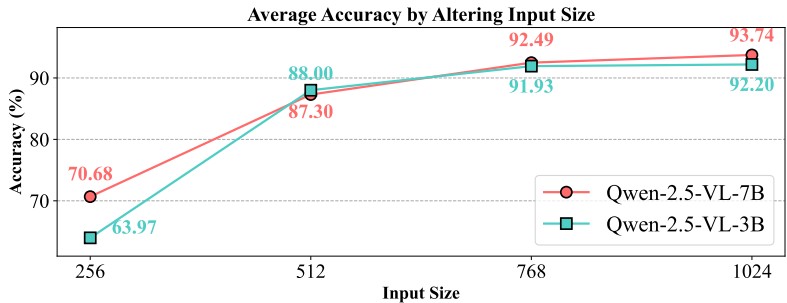

Figure 4: Performance (Acc (%)) across different input resolutions on GenImage.

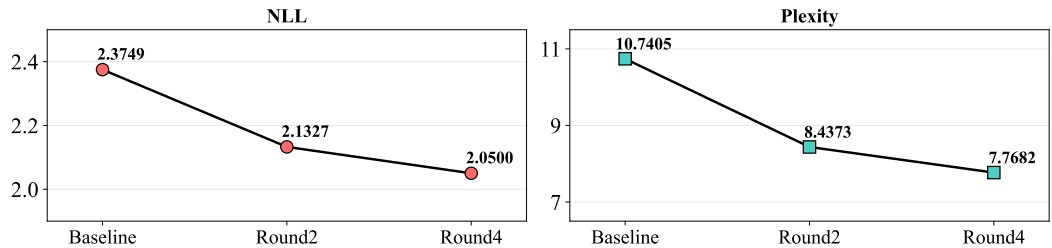

Figure 5: Impact of dialogue turns on data alignment with the pre-trained Qwen2.5-VL-7B. As the number of turns increases, both Negative Log Likelihood (NLL) and Perplexity decrease, suggesting multi-turn dialogues are more consistent with the model's inherent knowledge.

this hypothesis, we designed an experiment to measure how data format impacts alignment with a pre-trained model's knowledge. Starting with a base set of 400 annotations ($\mathcal{A}_{base}$), we generated three distinct formats: (1) a traditional single-turn pair, (2) a dialogue with 2 rounds, and (3) a dialogue with 4 rounds. The multi-round dialogues were generated by Gemini 2.5 pro from $\mathcal{A}_{base}$. Due to the requirement to split information across the dialogue, Gemini autonomously embeds additional user prompts to maintain a coherent context. We then measured the Negative Log Likelihood (NLL) and Perplexity of each format on a pre-trained Qwen-2.5-VL-7B model, with lower values indicating better alignment. As illustrated in Figure 5, the results are clear: both NLL and Perplexity decrease as the number of dialogue turns increases, even though the core information remains constant. This finding indicates that the multi-round format lowers the learning barrier for MLLMs on this task. We speculate that this is because the additional user prompts act as a scaffold, breaking down the complex task into a series of smaller, more digestible steps that better align with the model's autoregressive nature.

Table 8: Ablation study of our methods for different stages (ACC (%)).

| Variant | GenImage | | | GenImage++ | | | | AVG |
|---|---|---|---|---|---|---|---|---|
| | MidJourney | BigGAN | SDv1.5 | Flux | Flux Real | SD3 | SD3 Real | |
| Qwen-2.5-VL-3B | 83.23 | 94.58 | 97.81 | 99.57 | 99.92 | 96.30 | 98.83 | 95.61 |
| Qwen-2-VL-7B | 88.84 | 97.84 | 99.04 | 97.57 | 97.37 | 97.82 | 98.85 | 96.76 |
| Qwen-2.5-VL-7B | **93.20** | **98.49** | **98.85** | **99.58** | **99.97** | **97.88** | **99.78** | **98.25** |

**Ablation for different foundation models** To verify the universality of our method, we applied the proposed framework to different foundation models, including Qwen-2.5-VL-3B, Qwen-2.5-VL-7B, and Qwen-2-VL-3B. The results, shown in Table 8, demonstrate the consistent effectiveness of our approach across architectures.

**Evaluation on More Benchmarks** To comprehensively evaluate the effectiveness of Forensic-Chat, we further conduct experiments on several AIGI detection benchmarks. The results, reported in Tables 9, 10, and 11, strongly validate the effectiveness of our method and further highlight the outstanding performance of Forensic-Chat.

Table 9: Generalizable performance (ACC (%)) on EvalGen (Chen et al., 2025). We obtained the experimental results from the original paper. This dataset includes fake images only.

| Method | Flux | GoT | Infinity | OmiGen | NOVA | **AVG** |
|---|---|---|---|---|---|---|
| UnivFD | 4.00 | 9.20 | 15.70 | 8.30 | 39.60 | 15.40 |
| FatFormer | 9.90 | 47.80 | 44.70 | 98.30 | 27.30 | 45.60 |
| C2P-CLIP | 8.70 | 49.40 | 35.10 | 86.40 | 14.80 | 38.90 |
| AIDE | 16.20 | 21.60 | 4.00 | 14.90 | 18.40 | 15.00 |
| AlignedForensics | 45.00 | 84.40 | 79.60 | 90.80 | 85.20 | 77.00 |
| DDA | 87.00 | 99.30 | 99.50 | **99.50** | **100.00** | 94.00 |
| **Forensic-Chat** | **99.10** | **100.00** | **99.64** | 99.28 | **100.00** | **99.60** |

Table 10: Generalizable performance (ACC (%)) on evaluation set in Community Forensics (Park & Owens, 2025). The subset 'DALLE2' was ignored in this table.

| Method | DALLE3 | DFGAN | Flux-dev | GALIP | Hourglass | IdeogramV1 | IdeogramV2 | Imagen3 |
|---|---|---|---|---|---|---|---|---|
| C2P-CLIP | 64.15 | 99.30 | 60.25 | 74.44 | 68.70 | 51.45 | 50.80 | 50.38 |
| DeeCLIP | 91.00 | **99.85** | 57.70 | **86.35** | 65.75 | 67.55 | 62.15 | 78.43 |
| DRCT | 94.80 | 50.65 | 88.70 | 53.50 | 52.40 | 92.05 | 90.80 | 93.38 |
| PatchShuffle | **99.85** | 66.15 | 98.25 | 51.70 | 65.65 | 96.90 | 96.20 | 98.85 |
| **Forensic-Chat** | 99.05 | 99.65 | **99.80** | 84.30 | **98.45** | 96.75 | 96.50 | **99.05** |

| Method | Kandinsky | Kvikontent | LCM-SD15 | LCM-SDXL | LCM-SSD1B | MidJourney V5 | Stable Cascade | **AVG** |
|---|---|---|---|---|---|---|---|---|
| C2P-CLIP | 58.00 | 85.05 | 82.30 | 54.05 | 89.10 | 54.14 | 65.60 | 67.18 |
| DeeCLIP | 80.15 | 97.85 | 55.95 | 55.65 | 56.85 | 75.97 | 94.60 | 75.05 |
| DRCT | **99.75** | 98.65 | 95.85 | **97.90** | 89.85 | **97.82** | 97.90 | 86.27 |
| PatchShuffle | 99.50 | 99.30 | 92.55 | 71.30 | 58.70 | 81.51 | 94.40 | 84.72 |
| **Forensic-Chat** | 99.70 | **99.60** | **99.80** | 90.75 | **95.10** | 96.96 | **99.80** | **97.02** |

Table 11: Generalizable performance (ACC (%)) on evaluation set in SynthBuster (Bammey, 2024). The methods marked with * indicate results reported in the original paper. This dataset includes fake images only.

| Method | Glide | SD1.3 | SD1.4 | SD2 | SD XL | MidJourney | DALLE2 | DALLE3 | Firefly | **AVG** |
|---|---|---|---|---|---|---|---|---|---|---|
| UniFD* | 10.10 | 24.30 | 21.80 | 34.40 | 21.50 | 0.00 | 42.40 | 0.00 | 61.70 | 14.30 |
| C2P-CLIP | 12.00 | 51.10 | 54.20 | 39.10 | 56.20 | 6.50 | 12.00 | 27.20 | 19.70 | 30.89 |
| DeeCLIP | 48.30 | 93.30 | 93.40 | 68.16 | 54.10 | 30.00 | 89.44 | 0.40 | **71.50** | 60.96 |
| DRCT | 14.10 | 89.60 | 88.20 | **99.90** | 89.60 | **99.40** | 4.10 | 35.60 | 11.40 | 59.10 |
| PatchShuffle | 80.40 | 100.00 | 100.00 | 82.49 | 77.70 | 82.80 | 19.80 | 8.10 | 13.50 | 62.75 |
| **Forensic-Chat** | **99.90** | **99.10** | **99.10** | 37.00 | 22.80 | 44.60 | **99.8** | **96.10** | 32.60 | **70.11** |

### D.1 Visualization Results

We provide several visualization results of Forensic-Chat during image detection to illustrate how the model behaves in practical usage scenarios. In these examples, the proposed method is able to flexibly and naturally respond to diverse user instructions, demonstrating not only binary classification ability but also rich conversational competence. Thanks to the effective preservation of pre-trained knowledge during fine-tuning, Forensic-Chat can follow complex prompts, adapt its explanations to different levels of detail, and systematically organize its reasoning process. Moreover, it exhibits strong detection capability by capturing subtle and localized artifacts in fake images, such as distorted anatomy, inconsistent textures, or implausible lighting. At the same time, for real images, the model is able to provide positive evidence of authenticity—such as coherent environmental context, natural reflections, and physically plausible object interactions. Importantly, the explanations are not generic but targeted to specific visual regions, helping users understand the rationale behind the final verdict. These visualizations highlight the practical value of Forensic-Chat in real-world forensic scenarios, showing that it can serve as both a reliable detector and an interpretable assistant for human users.

---

**Real Example of Forensic-Chat**

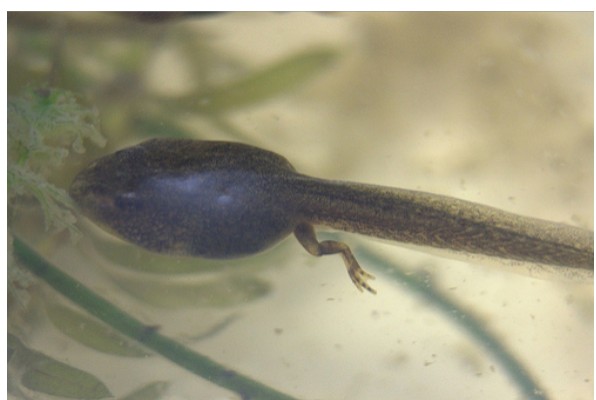

---

👤 **User:** Analyze the authenticity of this image. Please provide your analysis in a point-by-point list under the heading 'Authenticity Analysis'. Following the list, provide a final verdict under the heading 'Conclusion'.

🤖 **Forensic-Chat:** Of course. Here is the authenticity analysis for the provided image.

**Authenticity Analysis:**

- *Lighting and Reflections*: The lighting in the image appears natural and consistent with an underwater or semi-aquatic environment. The reflections on the water's surface accurately capture the shape of the tadpole, which would not be the case if it were digitally added.

- *Morphological Consistency*: The proportions of the tadpole's head to its body are anatomically correct for a young frog. While they may appear unusual from some viewing angles, this is a natural stage in the life cycle, not a sign of manipulation.

- *Environmental Context*: The tadpole is resting on what looks like aquatic vegetation or moss. This detail places the subject firmly within a logical and realistic ecosystem. There are no out-of-place elements to suggest it was artificially inserted.

**Conclusion:** Based on the consistent lighting, anatomical correctness, and coherent environmental context, the image is assessed as **authentic and shows no signs of being AI-generated or digitally manipulated**.

---

**Fake Example of Forensic-Chat**

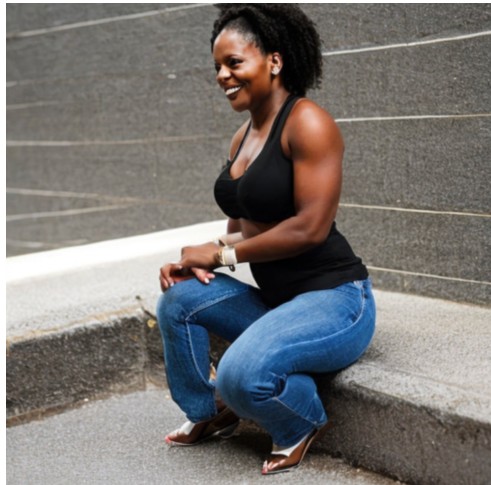

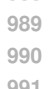 **User:** Can you describe this photo?

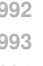 **Forensic-Chat:** This is a photograph of an African-American woman with dark curly hair, smiling at the camera. She has on a black tank top and blue jeans, and she is crouched on what appears to be a low concrete curb or wall. The background consists of a gray stone wall.

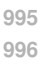 **User:** A Tell me your analysis

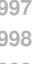 **Forensic-Chat:** Certainly. Here is a more detailed breakdown of the inconsistencies I found:
**Face:** The woman's teeth appear unnaturally blurred, and her eyes lack a clear pupil and iris detail.
**Hands:** The fingers are misshapen and appear distorted upon closer inspection.
**Body Shape:** There is an unusual asymmetry between her left.
**Accessories:** The watch on her wrist seems to merge into her skin.

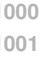 **User:** I think there are some issues with her ankle

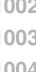 **Forensic-Chat:** That's correct. Upon examining the lower part of the image where her legs meet the ground, the heel of the shoe appears unnaturally flat and merges into the surface without a distinct transition, which further supports the conclusion that the image is likely AI-generated.

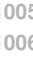 **User:** Anything else?

**Forensic-Chat:** Notable artifacts also exist in the background. The lines defining the concrete steps behind her are not straight and show signs of distortion, which deviates from how such surfaces would typically appear in a real-world setting.

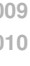 **User:** Tell me more details for this artifacts

**Forensic-Chat:** In the background, the lines of the concrete steps are irregular. They don't maintain a perfectly parallel alignment or consistent thickness throughout their length. This kind of geometric inconsistency is a common artifact in AI-generated content that struggles to render repeating patterns with perfect accuracy, even when they are simple elements like a stone wall.

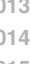 **User:** Systematically organize your responses, analyze them point by point, and provide the final conclusion

**Forensic-Chat:**
* **Facial Features:** Discrepancies include blurred teeth, indistinct eye structures (no visible pupils or irises), and an unnatural symmetry in the mouth area.
* **Hair:** The hair lacks fine texture and appears too uniform.
* **Hands:** The fingers are malformed and poorly defined.
* **Body Shape:** Asymmetrical limbs, specifically the uneven appearance of the arms.
* **Accessories:** An accessory like a watch merges unnaturally with the skin, lacking a clear boundary.
* **Clothing:** The clothing does not accurately represent physical folds and creases; instead, it appears rigid and flattened.
* **Background:** Inconsistent and distorted lines on the stone wall in the backdrop.

**Conclusion:** Based on the accumulation of these specific artifacts—particularly the distorted hands, unnatural facial features, and illogical merging of objects—the image is highly likely to **be AI-generated and not a photograph of a real person**.

# E  DATASET CONSTRUCTION

In this section, we introduce the details of our dataset $\mathcal{P}_1$ and $\mathcal{P}_2$.

## E.1  DATASET $\mathcal{P}_1$ IN STAGE 1

**Image Source**  In Stage 1, we selected 25,000 real images from the MS-COCO dataset and used a pre-trained VAE (SD 2.1) to generate pseudo-fake counterparts. The VAE compresses each real image into the latent space and then reconstructs it back into the pixel space, a process that introduces low-level artifacts without introducing significant semantic errors. In total, we constructed a dataset of 50,000 images, consisting of 25,000 real and 25,000 pseudo-fake samples.

**Instruction Fine-Tuning Data**  In Stage 1, our method aims to enhacne the visual perception to AI-generated images of MLLMs. To this end, we constructed a minimalistic instruction-following dataset guided by two core principles. First, the instruction-response pairs are designed to have a high joint probability within the pre-trained model's learned distribution. This strategy aims to minimize perturbations to the model's original representation space, thereby preserving its general capabilities. Second, the instructions are intentionally made concise and free of extraneous semantic information. This ensures that the model's learning is focused squarely on the visual perception task of distinguishing between real and AI-generated content, rather than on interpreting complex commands. Consequently, we employed direct, closed-ended questions that solicit single-word answers, compelling the model to ground its judgment in visual evidence. This targeted fine-tuning approach effectively enhances the model's sensitivity to AI-generated artifacts while safeguarding its foundational knowledge. We provide two representative examples below.

---

**Example 1 ($\mathcal{P}_1$)**

**Instruction:** Please determine if the following image is real or fake. You can detailed check the image and find the evidence. Please just answer 'real' or 'fake'.
**Response:** real

---

**Example 2 ($\mathcal{P}_1$)**

**Instruction:** After analyzing this image, are there clear signs of AI generation? Answer 'yes' or 'no'.
**Response:** yes

---

## E.2  DATASET $\mathcal{P}_2$ IN STAGE 2

**Image Source**  The training dataset for Stage 2 comprises 34,000 images, balanced equally between real and AI-generated examples. The 17,000 real images are randomly sampled from Unsplash (Unsplash, 2025), Hand-Body (Narasimhaswamy et al., 2022), and LAION (Schuhmann et al., 2022). The corresponding 17,000 AI-generated images are sourced from the GenImage Zhu et al. (2023) (SDv1.4) training set (5,000 images), SynthScars (Kang et al., 2025) (5,000 images), and Echo-4o (Ye et al., 2025) (250 images), supplemented by 6,750 images that we generated using the Flux.1-dev Labs (2024).

**Instruction Fine-Tuning Data**  As part of our **Dialectical Fine-Tuning (DFT)** in Stage 2, we construct a semantic-artifact dataset, denoted as $\mathcal{P}_2$. This process is designed to generate data that explicitly models the **contradiction between visual evidence and commonsense knowledge**, thereby fostering the model's reasoning capabilities. The data generation pipeline consists of three main steps:

1. **Step 1: Extracting Visual Evidence.** For each image, we first use Gemini-2.5-Pro to generate a detailed caption to its visual content. This description, which we denote as $c$, serves as the **visual evidence** ("what it sees"), factually capturing any semantic artifacts or anomalies present in the image. To generate this visual evidence $c$, we input the prompt $V_1$ into Gemini-2.5-Pro.

2. **Step 2: Formulating a Commonsense Counterpart.** Subsequently, we again leverage Gemini-2.5-Pro to perform a semantic inversion on the initial caption $c$. The goal is to produce a corresponding **commonsense rule**, denoted as $c^r$, which represents the model's **internal world knowledge** ("what it knows"). This inverted caption is not a mere negation but a statement of fact that creates a direct dialectical conflict with the anomalous visual evidence. For instance, if the visual evidence $c$ is *"The figure in the image has six fingers,"* the corresponding commonsense rule $c^r$ would be *"A normal human has five fingers."* To generate the commonsense rule $c^r$, we input the prompt $V_2$ into Gemini-2.5-Pro.

3. **Step 3: Synthesizing Multi-Turn Dialogues.** Finally, the pair $\{c, c^r\}$, which encapsulates the core contradiction, serves as a seed annotation. To avoid the pitfalls of single-turn and fixed format instruction tuning, which can encourage shortcut learning, we transform this seed annotation into a **multi-turn dialogue** with random rounds (1-4). This conversational format decomposes the complex reasoning task into progressive, context-reasonable steps, aligning better with the model's autoregressive pretraining. The resulting dialogue explores the conflict between the visual evidence ($c$) and the commonsense rule ($c^r$), effectively disentangling *what* to reason about from *how* to present the answer. To synthesize each multi-turn dialogue, we use Gemini-2.5-Pro as the Large Language Model and input the prompt $V_3$.

We exhibit the prompts $V_1$, $V_2$, and $V_3$ below, along with some examples of training data.

---

**Prompt $V_1$ (Input: `LABEL`, `Image`) (Part 1)**

---

*System:* You are an image-forensics expert whose sole task is to determine whether a given picture is a genuine photograph or an AI-generated creation. Leverage every analytical tool at your disposal and reason rigorously, examining each aspect of the image for tell-tale artifacts or authentic cues. Draw on common sense, domain knowledge, and real-world experience to deliver a clear, comprehensive, and accurate assessment—and explain your verdict step by step.'

*Prompt:* This is a {`LABEL`} image. Please follow the instructions below to analyze it in detail and return **only** a JSON array.

## Analysis dimensions (scan each dimension thoroughly)

1. **Scene realism**
- Does the scene belong to the physical world?
- Is the image surreal / 3-D rendered / impossible in real life (e.g., cinematic lighting, fantasy architecture, highly stylized or painterly rendering)

2. **Object defects & anomalies** (including but not limited to)
- Shape distortion / breaks / holes
- Unnatural texture repetition, stretching, or floating pixels
- Perspective or proportion errors

3. **Lighting & shadows**
- Are light direction, shadows, reflections, and exposure natural and self-consistent?

4. **Focus & depth of field**
- Are foreground/background blur and edge fall-off appropriate?

5. **Sharpness consistency**
- Are resolution and noise distribution uniform across the frame?
- Are some areas extremely detailed while others are overly smooth?

6. **Object interactions**
- Are occlusion, contact, and cast shadows between multiple objects reasonable?

7. **AI texture artifacts**
- Brush-stroke patterns, noise, over-smoothing, smearing, etc.

8. **Stylistic clues**
- AI-generated images often share characteristic styles or compositions.
- Decide whether this image matches a typical AI style or filter.

Below is a clear, hierarchical, and logically complete checklist of artifact inspection points. You may refer to the following checklist, but it is not limited by it.

---

**Prompt $V_1$ (Input: `LABEL`, `Image`) (Part 2)**

**1 Geometric and Structural Consistency**
- **Perspective & Lighting**: Shadow direction conflicts, depth distortion, mirror reflections not matching single light source models
- **Physical Details**: Incorrect cloth folding logic, wrong glass refraction angles
- **Biological Structures**: Extra/missing fingers, mismatched earrings, incorrect number of animal limbs/claws
- **Text Accuracy**: Collapsed or jumbled signboards/road signs, especially in multi-line text
- **Edges & Seams**: Edge drifting, excessive hair-background blending, sudden breaks in continuous areas
- **Smudging in Complex Areas**: Large area blurring to avoid complex structures (e.g., crowds, leaves)

**2 Semantic and Common Sense Consistency**
- **Scene Logic**: Rainbow in night sky, stars with direct sunlight
- **Fantasy / Unreal Elements**: Magical castles, dogs piloting planes, screens growing out of animals
- **Over-Idealization**: Flawless faces, extreme symmetry, overly high saturation
- **Repeated Textures**: Looping patterns on floor tiles, lawns
- **Uniform Micro-Expressions**: Everyone in group photos showing the same exact expression
- **Abnormal Object Interactions**: Violations of physical rules like interpenetration

**3 Indoor Scene (Room, etc.) Artifact Check**
- **Structural Integrity**: Walls, doors, and windows must be naturally connected and closed; no fractures or missing parts
- **Spatial Logic**: Avoid floating, clipping (through walls/tables/bathtubs, etc.)
- **Mirror Reflections**: Reflected content matches real space with consistent perspective
- **Materials & Textures**: No texture stretching, misalignment, or abrupt seams
- **Perspective Consistency**: Parallel lines (wall corners, floor tiles) converge to the same vanishing point; avoid dual perspectives
- **Semantic Consistency**: Furniture size proportion, walking paths, and functional layout should be reasonable
- **Lighting & Shadows**: Light source positions, shadow directions, and intensities must be natural and consistent

**4 Human-Related Artifacts**
- **Eyes**: Size, color, or highlights mismatch between left and right; distorted shapes
- **Teeth**: Missing edges, blurry blending, overly smooth
- **Ears / Accessories**: Left-right size or position deviations; mismatched earrings; glasses not fitting the face
- **Hair**: Texture distortion, missing patches, or floating against gravity
- **Hands / Body**: Finger/limb deformities, overly uniform facial features in multiple people
- **Background Characters**: Missing facial details, strange expressions; incorrect shapes of held objects (e.g., cameras)

**5 Outdoor Scene (Architecture • Landscape, etc.) Artifacts**
- **Structural Integrity**: Buildings, roads missing or deformed
- **Spatial Logic**: Floating, sinking into ground, or "far object blocking near object"
- **Occlusion Relationships**: Reversal of depth layers
- **Materials & Textures**: Texture stretching, repeating patterns, mosaics, misaligned stitching
- **Perspective Consistency**: Single vanishing point; avoid conflicts from distortion or multiple vanishing points
- **Semantic Consistency**: Proper scale and realistic combinations (e.g., grass not growing on rooftops)
- **Lighting & Shadows**: Unified direction and intensity

**6 Target Objects (Animals • Vehicles • Food, etc.) Artifacts - Symmetry**: Unequal eye sizes, deformed oval tires
- **Edge Transition**: Blurry or unclear boundaries blending into background
- **Icons / Text**: Blurred or distorted license plates, package labels
- **Structural Logic**: Bent shapes, hollow/solid errors
- **Component Integrity**: Missing guitar headstock, mouse with fewer claws
- **Shadows & Reflections**: Missing shadows despite consistent lighting, or wrong shadow directions
- **Object Interactions**: Tire marks not aligning with ground
- **Unreal Objects**: Absurd structures like bread used as wheels
- **Background Issues**: Oddly shaped doors/windows, perspective errors

**Prompt $V_1$ (Input: `LABEL`, `Image`) (Part 3)**

**## Output requirements**
- If the whole image is surreal / 3-D rendered / impossible in real life (e.g., cinematic lighting, fantasy architecture, highly stylized or painterly rendering), must state that first.
- You may refer to the analytical dimensions mentioned above, but do not restrict yourself to them. These clues may or may not appear in the image; your analysis should be grounded in your professional knowledge and experience, and should be comprehensive based on the actual content of the image.
- Identify every anomaly or noteworthy normal feature and write one analysis entry per finding, using the dimensions above or your own perspective.
- Scan foreground, mid-ground, and background so nothing is missed.
- When you reference a local region, crop it and provide normalized coordinates in bbox2d as ( y min, x min, y max, x max ) (values in [0, 1000]).
- If the description concerns the entire image, set 'bbox2d' to an empty list [].
- The bounding-box coordinates must be exact, align precisely with the referenced image region, and you must pay extremely close attention to meeting this requirement.
- Also record plausible regions—do not skip parts that look correct. In fake images, do not overlook regions that appear plausible; note that they look reasonable, but avoid making any definitive conclusions about them.
- Tailor your reasoning to the different condition. For example:
- If a region is tiny, say so before analyzing it.
- If details are unclear, lower your confidence accordingly.
- For each analysis entry, first state the region and its condition (status), then—integrating that status—perform the artifact analysis. For example: "In the top-left corner, there is a small object that appears to be a tree. It looks unnatural because . . . However, it might not be a confident clue since I cannot see it clearly."
- Explain as if you were talking to a user who has *not* seen this prompt.
- Keep the language plain and easy to understand.
Important Things to Note: YOU MUST NOT OUTPUT THE POINT IF YOU ARE NOT VERY CONFIDENT ABOUT! CONTROL YOUR ANALYSIS TO ONLY THOSE YOU ARE VERY CONFIDENT ABOUT!

**Prompt $V_2$ (Input: LABEL, Image, DESCRIPTION $c$)**

***System:*** You are an image-forensics expert whose sole task is to determine whether a given picture is a genuine photograph or an AI-generated creation. Leverage every analytical tool at your disposal and reason rigorously, examining each aspect of the image for tell-tale artifacts or authentic cues. Draw on common sense, domain knowledge, and real-world experience to deliver a clear, comprehensive, and accurate assessment—and explain your verdict step by step.'

***Prompt:*** This is a description to a {**LABEL**} image. You should carefully extract the key information of this description and then process every object as follows:
- If the input image is fake or AI-generated, describing how the object should appear in reality.
- If the input image is real or natural, describing how the object should appear in ai-generated.
The description you should process is:
{**DESCRIPTION**}

---

**Prompt $V_3$ (Input: LABEL, Image, SEED ANNOTATION $\{c, c_r\}$)**

***System:*** You are a helpful assistant.
***Prompt:*** You are an expert in dialogue and image analysis. Your task is to construct a coherent multi-turn conversation based on the **provided information** and the **predefined scenario**. The conversation is between a user and a multimodal large language model (MLLM). The user's tone should be objective and direct, focusing on questions without unnecessary embellishments. You may imagine plausible user questions based on the scenario. The user's questions and the expert's answers must be consistent and reasonable, meaning the expert must respond according to the user's question, and the user's question should cover the scope of the expert's response. For example, if the user only asks for a description of the image content without mentioning authenticity analysis, the expert should not provide authenticity analysis but only describe the image content. You may polish the provided information to make the responses more natural and reasonable, but the dialogue content must be based on the **provided information**, and must not include any **fabricated information**. The multi-turn dialogue should be logically structured and coherent, with each turn naturally building upon the previous one. The user's question may refer to only a part of the content, and the expert's answer should provide a detailed analysis of that part. The user may also ask about all content at once, and the expert's response should cover the complete analysis of the entire image. In the conversation, only the first-round question is initiated spontaneously by the user. All subsequent questions must be based on the content of the expert's previous answer. In addition, the user does not know the content or authenticity of the image beforehand, so the user's first question should be open-ended.

---

The scenario is set as follows:
{**SCENARIO**}

---

You should generate the dialogue by the **provided information** as follows:
{**SEED ANNOTATION**}

**Example 1 ($\mathcal{P}_2$)**

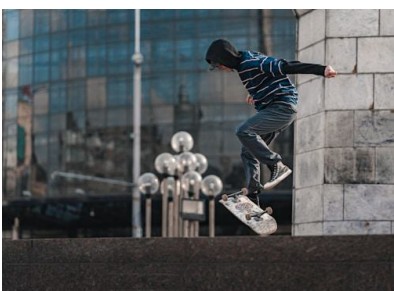

**Instruction:** Analyze the authenticity of this image. I need a point-by-point analysis and a final conclusion. For each point, please identify the clue, explain why you think it's real or fake, and then provide a counter-analysis for the opposite case. Use the following format for your response:

**[Clue 1]**
- [Why real / Why fake]: [reason]
- [If fake / If real]: [reason]
**[Clue 2]**
- [Why real / Why fake]: [reason]
- [If fake / If real]: [reason]
...
Conclusion: This is a [real / fake] image

**Response:** Of course, here is a point-by-point authenticity analysis of the image as you requested.
**1. Lighting and Shadows**
- *Why real*: The lighting in the scene is consistent and behaves according to the laws of physics. The highlights on the skateboarder and the shadows cast by him and the skateboard are coherent with a single, diffused light source from above and to the left, which is typical of an overcast day. The soft shadows on the ground and wall are physically plausible.
- *If fake*: If this were an AI-generated or composite image, there might be lighting inconsistencies. For example, the skateboarder could have highlights from a different direction than the ambient light, or the shadows might be too sharp, too faint, or pointing in the wrong direction.
**2. Motion and Focus**
- *Why real*: The image exhibits natural depth of field and motion blur. The skateboarder, as the main subject, is in sharp focus, while the background is blurred (an effect called bokeh), which is characteristic of a camera lens with a wide aperture. There is also a slight, natural motion blur on the skateboard, indicating it was captured mid-action with a fast shutter speed.
- *If fake*: An AI-generated blur might create unnatural artifacts, such as a sharp, cutout-like edge around the subject, or apply a uniform blur to the entire background without accounting for depth. A fake motion blur might look like a simple filter rather than a natural effect tied to the object's speed.
**3. Reflections and Textures**
- *Why real*: The details in the reflections on the glass building and the chrome spheres are complex and consistent. The windows reflect parts of the surrounding environment in a distorted but physically correct manner. The textures on the clothing, the worn skateboard deck, and the stone walls all show realistic detail and imperfections.
- *If fake*: AI models often struggle with creating coherent, complex reflections. The reflections might appear nonsensical, distorted incorrectly, or inconsistent with the visible environment. Textures might look too perfect or repetitive.
**4. Subject and Pose**
- *Why real*: The skateboarder's posture, balance, and interaction with the skateboard are authentic to the action being performed. The way his body is tensed, his arms are out for balance, and his feet are positioned on the board is a natural and dynamic pose for a skateboard trick.
- *If fake*: An AI-generated human figure might have subtle anatomical errors, such as unnatural limb positions, incorrect joint bending, or an awkward, weightless posture that doesn't align with the action being depicted.
**Conclusion**: This is a **real** image. The consistency of lighting, natural camera effects like depth of field and motion blur, and the complexity of reflections and textures strongly indicate that this is a genuine photograph of a moment captured in time.

**Example 2 ($\mathcal{P}_2$)**

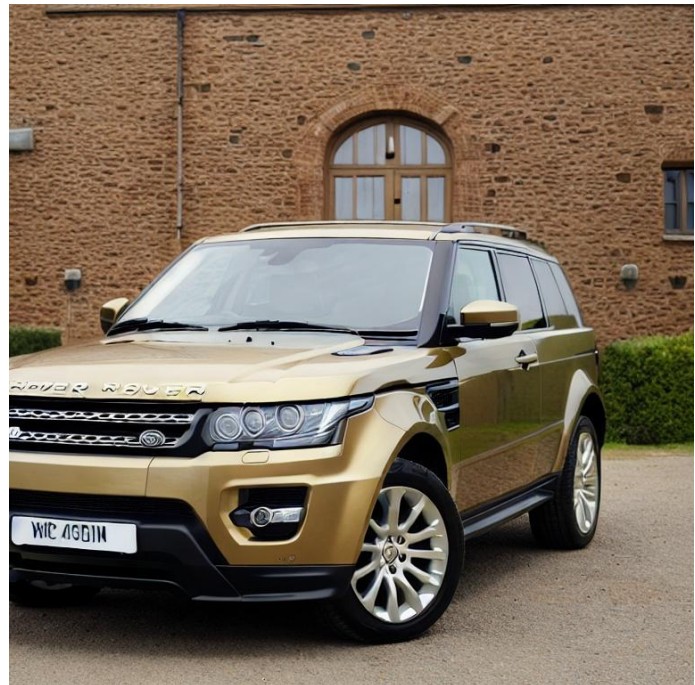

**Instruction:** What is depicted in this image?

**Response:** This image shows a gold-colored SUV parked on a paved surface. The vehicle appears to be a Range Rover model. In the background, there is a large building with a brick facade, an arched window, and some green bushes to the right side of the frame.

**Instruction:** Analyze the authenticity of this image. Please provide your analysis in a point-by-point format. After the points, give a final conclusion using the template 'Conclusion: [Your conclusion here]'.
**Response:** Here is the authenticity analysis of the image:
* **Wheels:** The edges of the car wheels are unclear and lack sharpness, whereas they should be well-defined in a real photograph.
* **License Plate:** The license plate is blurred, and the text on it is indistinguishable. In a genuine photo, this text should be clear and legible.
* **Emblem:** The car emblem on the grille lacks discernible contours and a defined shape, which is inconsistent with a real vehicle's branding.
* **Headlights:** The small lights within the headlight assembly are distorted and do not have a smooth, regular shape as they would on a manufactured car.
**Conclusion:** Based on these inconsistencies, it is highly likely that this image is AI-generated.

**Instruction:** You mentioned the license plate and the emblem. Can you elaborate on the distorted text on the vehicle, specifically on the hood and the license plate?
**Response:** Certainly. The text on the front edge of the hood, where 'RANGE ROVER' should be clearly visible, appears as a series of jumbled and misshapen characters that are not legible. Similarly, the text on the front license plate is completely blurred and unintelligible, failing to form any recognizable letters or numbers. In an authentic image, both of these text elements would be crisp and readable.

### E.3 EXPLAINFAKE-BENCH.

In this paper, we introduce a new benchmark, ExplainFake-Bench, designed to evaluate the explainability of MLLM-generated outputs in AI-generated image detection. The dataset is constructed by randomly sampling 400 fake images each from Chemeleon (Yan et al., 2024a) and GenImage++ (Zhou et al., 2025a), resulting in a total of 800 samples. For each image, we assign a randomly selected instruction phrased in different ways to query its authenticity, and then collect explanatory responses generated by MLLMs. To assess the quality of these explanations, we adopt the LLM-as-Judge framework (GPT-4o) and evaluate along five dimensions: (1) **Correctness**, which ensures that the final judgment aligns with the ground-truth label; (2) **Specificity**, which measures whether the explanation refers to concrete visual details rather than vague statements; (3) **Logical Consistency**, which assesses whether the reasoning is coherent and free from contradictions; (4) **Factual Accuracy**, which checks whether the cited details are consistent with the actual image content; and (5) **Instruction Following**, which evaluates adherence to the given instructions and required format. Together, these dimensions comprehensively measure both the validity and the explanatory quality of model outputs. The evaluation prompt is provided below.

---

**LLM-as-Judge (Input: `LABEL, IMAGE, DESCRIPTION`)**

*Prompt:*

You are an expert in evaluating the quality of model judgments. You will receive:

- A text description $D$ (this description claims whether the image is AI-generated or real, and provides supporting reasons)
- An image $I$
- A label $L$ ("Real" or "Fake")

Your task is to strictly evaluate $D$ according to the scoring dimensions below and output scores.

**[Evaluation Dimensions and Scoring]**

*1) Correctness (1–5 points)*

- Assess whether the final conclusion of D—classifying the image as "AI-generated" or "real" (whether explicit or implicit)—is consistent with the visible evidence in image I.
- If the conclusion is clearly contrary to the image, score 1; if no clear conclusion is given, <=2 points.

*2) Specificity (1–5 points)*

- Are the arguments specific, pointing to explicit regions/objects/relationships in the image, avoiding vague statements (e.g. "looks like AI")?

*3) Logical Consistency (1–5 points)*

- Narrative is consistent, causally reasonable, quantitative claims match conclusions; avoids contradictions, circular reasoning, irrelevant logic.
- 

*4) Factual Accuracy (1–5 points)*

- Are details in D consistent with the actual content of image I; avoid speculation, misidentification of objects/text/numbers/relationships.

*5) Instruction Following (1–5 points)*

- Does the description follow the task instructions properly, addressing all required elements and adhering to the requested format and focus?

**[Scoring Method]**

- If image quality is extremely poor/subject blocked: appropriately lower "Evidence Sufficiency/Specificity", not necessarily 1; if D explicitly acknowledges uncertainty and gives reasonable verification suggestions, partial credit may be retained.
- If the judgment result is wrong (inconsistent with label L): each dimension may still be scored, but it should be heavily penalized. (the score should be <=2).

**[Output Requirements]**

Output in the following format: <judgement>Your scoring rationale per dimension</judgement>
<scores>
[Correctness]: X
[Specificity]: X
[Logical Consistency]: X
[Factual Accuracy]: X
[Instruction Following]: X
</scores>
Where X is between 1–5.
The following needs to be evaluated:
### Text Description $D$:
{DESCRIPTION}
### Label $L$:
{LABEL}
### Image I:
{IMAGE}

---

