# OpenReview forum: "Seeing Before Reasoning: A Unified Framework for Generalizable and Explainable Fake Image Detection"
_ICLR.cc/2026/Conference — ICLR 2026 Conference Withdrawn Submission_

### Official Review · Reviewer_AYcS · 2025-10-22

**Soundness:** 3
**Presentation:** 4
**Contribution:** 2
**Rating:** 4
**Confidence:** 4

**Summary:**

The paper proposes a two-stage MLLM fine-tuning method for fake image detection. In the first stage, the method freezes the LLM and only finetunes the image encoder using autoregressive loss and yes/no or fake/real target (binary classification). In the second stage, the image encoder is frozen and the LLM is finetuned using image-text pairs. For the first stage, pseudo-fake images are generated using VAEs; the reconstructed image is used to train the image encoder, as it only includes fine-grained artefacts. For the second stage, Gemini 2.5 is used to generate the text descriptions based on what Gemini sees (eg an image of a truck with weird proportions) vs what it knows about the object (eg trucks are symmetric). Finally, the paper proposes a benchmark for the evaluation of MLLMs and their explainability capabilities.

**Strengths:**

The paper is very well written, and it was very easy to follow the motivation and method. The method is intuitive; the two-stage finetuning approach has been used previously in other tasks (BLIP, LlaVa in general MLLM) and, as expected, yields good results here as well.
The experiments are well executed and convincing, particularly as we see the cross-dataset performance.
In addition, the ablation study shows the importance of each component.

**Weaknesses:**

The main weaknesses are related to the motivation and experimental setup. More specifically:
- The authors use Gemini to annotate the data in stage 2, without human annotations. As such, the method at stage 2 is more of a distillation method (ie, distilling Gemini knowledge into Qwen on the downstream task of fake image detection)
- Related to this, it is hard to assess the usefulness of the method, as we have no context into how well Gemini performs (both in the dataset creation and also in the wild), as well as how well the base method performs without any training
- In terms of catastrophic forgetting, the way the paper is written, it looks like this is attributed to the two-stage training; however, avoiding catastrophic forgetting is a well-known property of LoRA adaptors (that are used here). As a result, the claim on catastrophic forgetting is overstated (LoRAs are an implementation detail, not a key methodological contribution), and to properly assess this claim, we would need to see how Forensic-Chat is performing when fully finetuned (without adapters).
- Human evaluation of the explanations is not included.
- The novelty of the two-staged approach is a little overstated, in the sense that it has been done in other tasks and is pretty standard for MLLMs, so it would be best to recognise this in the text.

Some minor weaknesses and errata:
- End-to-end training performance is not shown in the ablations
- Lines 300-301 are unclear. Without looking into the appendix, the reader simply does not understand what has happened there.
- Lines 121 & 125: "and etc" is added, but it seems that it's better to list exactly what is done, as this is the introduction, and the reader needs a clear understanding.
- In the list of contributions, the last bullet point is reiterating points 1 & 2.

I am willing to raise my rating if these concerns are addressed, although a key concern is whether there is novelty and even motivation to distil knowledge from a very large model (LLM) into a task-specific MLLM.

**Questions:**

- How does Gemini perform in terms of accuracy and explainability?
- How does Forensic-Chat perform on the general MLLM understanding (Table 7) when fully finetuned (NOT using LoRA)?
- How do descriptions generated by Gemini and ForensicChat generated descriptions resonate with human evaluators?

---

### Official Review · Reviewer_pX5C · 2025-10-25

**Soundness:** 2
**Presentation:** 1
**Contribution:** 3
**Rating:** 4
**Confidence:** 5

**Summary:**

The main contributions of this paper are in the following three aspects:
1. Model training. A two-stage training approach is proposed. In the first stage, only the visual encoder is trained to enhance its ability to perceive artifacts. In the second stage, high-quality multi-turn dialogues are used to improve the model’s fake reasoning capabilities.
2. Data annotation. A high-quality data annotation process tailored to the two-stage training process is designed and constructed.
3. Evaluation: A new evaluation benchmark, ExplainFake-Bench, is proposed.

**Strengths:**

The paper establishes a novel data construction pipeline and evaluation protocol, and the research work is relatively solid.

**Weaknesses:**

The paper has significant issues in its writing, as it fails to provide sufficient effective information within the limited space of the main text.
1. In Section 3 (Method), a large portion of the content still focuses on discussing the motivation of the method rather than the paper’s technical details. The training method in Section 3.1 (autoregressive loss + selective freezing and training) is technically quite common, yet it is discussed at great length. In contrast, the data engineering part in Section 3.2— a key technology that determines whether the model can achieve performance improvement—is not clearly described. Readers must refer to Appendix E to understand the paper’s key innovations.
2. As a new evaluation protocol, ExplainFake-Bench only lists five evaluation dimensions in the main text. Readers need to consult the appendix to understand the details of the scoring criteria.
3. Important experimental settings used in the study (e.g., model initialization, source of training data) are not clearly specified in the main text.

It is necessary to remind the authors that reviewers are under no obligation to read appendices. Unlike general technical reports, formally published and accepted papers must ensure that all important technical details are included in the main text, rather than over-relying on information added in appendices.

**Questions:**

1. Generalization. What is the source of the images in the training dataset used for reporting experimental results? I would like to know whether the performance reported in Tables 1–4 corresponds to intra-dataset or cross-dataset evaluations.
2. Lack of data ablation. The new data construction pipeline is the key to the model’s performance improvement. However, to prove the effectiveness of this pipeline—especially the special design of "visual evidence + corresponding commonsense rule" in dataset P2—it is necessary to construct a baseline dataset with a scale comparable to P2. In this baseline dataset, the seed annotations of visual evidence should be directly converted into multi-turn dialogues, and the model should be trained on this dataset as a baseline. This will help determine the source of the performance improvement.
3. Data engineering is crucial for improving the performance of current MLLMs. Do the authors have plans to open-source the training data?

---

### Official Review · Reviewer_Rb3x · 2025-10-31

**Soundness:** 2
**Presentation:** 1
**Contribution:** 2
**Rating:** 2
**Confidence:** 5

**Summary:**

The paper proposes Forensic-Chat, a two-stage pipeline for AIGI detection using MLLMs. Stage 1 fine-tunes only the vision encoder on reconstruction-style data to sensitize the model to low-level artifacts, while freezing the LLM. Stage 2 fine-tunes only the LLM on multi-turn dialogue data designed to promote “dialectical reasoning,” i.e., connecting perceived artifacts to commonsense expectations of what a real image should look like. The topic is timely and clearly relevant to a top-tier venue like ICLR. However, I think the current draft is not ready for acceptance. The core concerns are novelty compared with existing practice, incomplete grasp of previous related works based on MLLMs, underspecified experimental controls, and unclear and weak evaluation for explainability.

Please see details below.

**Strengths:**

1. This paper has a clear problem framing. The paper identifies two coupled failure modes of current MLLM-based AIGI detectors: (a) vision encoders optimized for semantics instead of forensic artifacts, and (b) instruction-style finetuning with templated Q/A both induce shortcut learning and cause catastrophic forgetting.

2. The training recipe composed of stages 1&2 is simple and modular.

3. The authors evaluate on GenImage, GenImage++, AIGI-Holmes, WildRF, and AIGI-Bench. They also compare general vision-language ability on public multimodal benchmarks.

4. Table 5 shows Baseline vs. Stage 1 vs. Stage 2 vs. Stage 1+2, and breaks out where each stage helps.

**Weaknesses:**

1. The paper claims to first propose an entirely new paradigm..., where the model should first perceive the artifacts so that the reasoning process is truly based on the seen cues. However, the two-stage schedule “tune perception module first, then align reasoning head” is very reminiscent of standard staged adaptation/adapter tuning for VLMs. This is an incremental design choice, not obviously a fundamentally “new paradigm.”

2. The proposed Dialectical Fine-Tuning in Stage 2 is described as constructing multi-turn dialogues that progressively ask for deeper analysis and counterfactual reflection on what a ‘real’ image should look like. This is more of a structured chain-of-thought data curation rather than a new learning algorithm. Besides, can the authors provide quantitative proof indicating the conversation is actually diaretical?

3. Stage 1 uses dataset P1, which consists of pairs (I_real, I_recon) where I_recon is reconstructed by a pretrained VAE decoder, producing “pseudo-fake” images. In this part, the model is trained to output tokens like “real” / “fake,” with only the vision encoder updated and the LLM frozen. Stage 2 uses dataset P2, where each image is annotated with multi-turn dialogues generated by an LLM.
What are the specific details, like size, diversity, or quality-control protocol, for P1 or P2? Besides, how is the quality controlled? Are there any human verifications? How did the hallucination in LLM-generated conversations get prevented?

4. The discussion attributes Stage2’s gain to “moving beyond reliance on fragile, low-level artifacts” and leveraging “robust features resilient to real-world distortions,” but does not quantify which distortions Stage2 is robust to.

5. The fatal flaw of this paper is the employed evaluation measures and the reported performance.
- First, there is no calibration/confidence reliability reporting. Since the method is presented as an interactive forensic assistant, its confidence statements must be trustworthy. Reporting only a generator-averaged accuracy sidesteps this completely. There is also no reliability diagram or analysis of whether high-confidence judgments are actually more trustworthy than low-confidence ones. This is concerning because a system that produces confident-sounding natural language explanations while being poorly calibrated is arguably more dangerous than a simple classifier: it can mislead users into over-trusting an incorrect decision. The metric does not meaningfully support the paper’s real-world generalization claims.

- Second, the reported “macro accuracy” is not standard. According to the results in the tables, it is the unweighted average of per-generator accuracies instead of the actual overall performance on each dataset, while generator subsets can themselves be class-imbalanced.

- Third, the paper argues strong robustness and “generalization to unseen generators / real-world distortions.”. But the reported score is a single scalar averaged across generators, with no visibility into which generators or conditions are actually weak.

6. Regarding the explanability aspects, there are also several concerns.
- The ExplainFake-Bench evaluation uses an “LLM-as-Judge framework,” with GPT-4o as evaluator, to rate each model along five dimensions, then average the scores. It is well-known that LLM judges show bias toward certain wording styles and safety disclaimers. This can systematically advantage models that share similar language patterns or formatting conventions. The paper does not acknowledge or control for this known bias. LLMs are used to generate the training data, and you also use LLMs to judge them. Wouldn't this cause a cycle-like paradox?

- There is no human evaluation or spot-audit to check factual grounding beyond automated scoring. There is no guarantee that the explanations faithfully correspond to actual image features, and the paper does not propose mechanisms to mitigate such hallucinations. In other words, what is the true clue or judgment to tell an image is fake or real? To what level does the model understand its visual artifacts for reasoning?

- Although text-level measures are employed, they only check textual similarity to pre-written ground-truth explanations, not whether the explanation is factually grounded in actual visual evidence. The model could hallucinate a plausible-sounding reason that matches reference wording but does not correspond to real image features. The evaluation does not include factual grounding metrics.

7. Why would the authors explicitly assume that the MLLMs detect AIGIs by visible visual cues instead of invisible patterns left by specific generators that are generally employed in conventional CNN-based detectors? Is there any empirical proof for this employed assumption? To my knowledge, the research on this aspect is still vacant now.

8. The authors argue that prior work either (a) co-fine-tunes vision+LLM and degrades general skills, or (b) uses an external expert detector and risks “just copying” that expert. However, I don’t see an ablation where you do co-fine-tune both vision and LLM end-to-end on P1+P2 and show that this harms general knowledge benchmarks dramatically more than Stage1+2. I either don’t see a teacher-distillation baseline.

9. The draft currently has noticeable grammatical issues and typos, such as subtal, first percept the artifacts, still-conversational assistant, etc. Some claims are hard to parse (“difficulty mismatch — simple prompts paired with elaborate answers — that encourages shortcut learning”). I highly suspect it as a cue of heavy LLM usage.

10. Stage 2 is referred to as “Dialectical Fine-Tuning (DFT),” and later you call it “Domain-Following Tuning (DFT)” in the ExplainFake-Bench discussion. What is the standard naming on earth? This is very confusing.

**Questions:**

Please see the weakness above.

---

### Official Review · Reviewer_9LRS · 2025-11-01

**Soundness:** 1
**Presentation:** 3
**Contribution:** 1
**Rating:** 2
**Confidence:** 5

**Summary:**

The paper introduces Forensic-Chat and ExplainFake-Bench. Forensic-Chat is a MLLM-based DeepFake reasoning model that makes use the ExplainFake-Bench data for supervised fine-tuning. ExplainFake-bench was curated by generating explanations from large MLLMs (proprietary models) like Gemini-2.5-Pro.

**Strengths:**

- The paper attempts to address a very real threat to modern media: protection against DeepFakes, which moves beyond binary real vs. fake classification and provides reasoning as to why the model thinks the provided image is real or fake.
- ExplainFake-Bench is a good attempt at making reasoning data publicly available for future models to train on. However, the scale of the data (~800 fake images) and the method used for generating the explanations at this scale is largely insufficient.

**Weaknesses:**

- The paper claims that Forensic-Chat is actually seeing before reasoning and not using "shortcut learning". However, there are no qualitative evidence for the same. For example, attention heatmaps to show that when the model is talking about the face or the hands, the attention heatmap actually points to those features of the image.
- The paper claims to be able to detect subtle anomalies within the images, however, the evaluations are only on fully AI-generated images (generated using models like Midjourney, SD, etc.) There are no experiments performed on face-manipulated data like DD-VQA, where the images originate from FaceForensics++ with manipulations only in the face, and the rest of the image being real. Most methods that work with FF++ crop the face as a preprocessing step, assuming a frontal human face is available. But when the full frame is passed, it would be important to check whether the Forensic-Chat model is able to look past the real components in the images (like the background, hands, etc.) and only focus and analyze the face. Only then can Forensic-Chat be called a truly generalized reasoning model.
- The two-stage training approach used does not actually ensure that the model is "seeing before reasoning". The visual encoder is tuned in stage-1 and LLM is tuned in stage-2, which essentially boils down to simple SFT techniques. Unless specific loss functions are devised to force the MLLM to ground its reasoning in visual cues the contribution in the architecture/method is extremely limited. As correctly pointed out by the paper, MLLMs indeed are built for high level semantics, and cannot capture subtle clues present in DeepFake images, however, such a method for training is not helping the MLLM. Intelligent design of the data protocol of each stage isn't enough to make a pretrained MLLM suddenly look for subtle clues and perform perfectly generalized DeepFake detection and reasoning.
- The evaluation of explainability relies on a new benchmark, ExplainFake-Bench, which uses GPT-4o as a judge. This "LLM-as-a-Judge" approach is inherently subjective and may carry the biases of the evaluator model, making the explainability scores difficult to verify independently. Accuracy has been the primary metric for comparison in the paper, which does not, at all, tell us whether the reasoning generated are meaningful or logical or hallucinated. Even NLG metrics like BLEU, ROUGE, etc. were not used which are popularly used in previous methods for DeepFake explainability. This makes it impossible to verify the claim that the model indeed provides good reasoning especially in cross-data or cross-generator settings.

**Questions:**

- Could the authors provide a more detailed analysis of the model's failures, particularly for the generators where it performed poorly (e.g., SD XL, Firefly)? What specific image characteristics or artifact types does the model struggle to detect?
- If MLLMs are actually as bad in subtle manipulation detection, why has the benchmark dataset been constructed using another (albeit bigger) MLLM like Gemini-2.5-Pro (which is a closed source proprietary model). And if models like Gemini-2.5-Pro indeed is able to generate good reasoning/explanations annotations for real and fake images, why would one need the Forensic-Chat framework to reason on DeepFakes? And if Gemini-2.5-Pro isn't actually generating good explanations, it makes the data to train Forensic-Chat significantly noisy, which again isn't helping the model learn meaningful features.

---

### Note · Authors · 2025-11-20

I have read and agree with the venue's withdrawal policy on behalf of myself and my co-authors.